# Full-Scale Maneuvering Trials Correction and Motion Modelling Based on Actual Sea and Weather Conditions

**DOI:** 10.3390/s20143963

**Published:** 2020-07-16

**Authors:** Bin Mei, Licheng Sun, Guoyou Shi

**Affiliations:** 1Navigation College, Dalian Maritime University, Dalian 116026, China; navi_captain@foxmail.com; 2Collaborative Innovation Research Institute of Autonomous Ship, Dalian Maritime University, Dalian 116026, China; allenimitsg@163.com

**Keywords:** full-scale maneuvering, trials correction, motion modeling, actual sea and weather conditions, reference model and support vector machine (RM-SVM), standards for ship maneuverability

## Abstract

Aiming at the poor accuracy and difficult verification of maneuver modeling induced by the wind, waves and sea surface currents in the actual sea, a novel sea trials correction method for ship maneuvering is proposed. The wind and wave drift forces are calculated according to the measurement data. Based on the steady turning hypothesis and pattern search algorithm, the adjustment parameters of wind, wave and sea surface currents were solved, the drift distances and drift velocities of wind, waves and sea surface currents were calculated and the track and velocity data of the experiment were corrected. The hydrodynamic coefficients were identified by the test data and the ship maneuvering motion model was established. The results show that the corrected data were more accurate than log data, the hydrodynamic coefficients can be completely identified, the prediction accuracy of the advance and tactical diameters were 93% and 97% and the prediction of the maneuvering model was accurate. Numerical cases verify the correction method and full-scale maneuvering model. The turning circle advance and tactical diameter satisfy the standards of the ship maneuverability of International Maritime Organization (IMO).

## 1. Introduction

During sea trial, ship motions include maneuvering and drifting. Drift motion is caused by the wind and waves at sea, and the ship shows slow, long periods of movement and even steady movement [1]. In order to obtain accurate trial data, the correction of this is an important step for ship maneuvering modeling. Dating back to 1978, Abkowitz utilized Esso Osaka for sea trials, identified the ship maneuvering mathematical model and verified the feasibility of the identification modeling method [2]. Recently, Zhang et al. [3], Bai et al. [4] and Kim et al. [5] also used full-scale ship data for identification modeling. In the literature [2,3,4], it should be noted that the log has also been installed underwater, on the ship hull, which is prone to suffering from cross flow, in addition to the ship being affected by the drift forces of wind and wave. Kim et al. [5] employed the method seen in the literature [6,7,8] to correct the sea trial data and identified the ship maneuvering model, but did not consider the influences of wind and waves. Using trial data to establish a model, one should choose the small-influence trials; otherwise, the influences of wind and waves need to be eliminated.

The International Maritime Organization(IMO) explanations for maneuvering standards [6], Society of Naval Architects and Marine Engineers(SNAME) guidelines [7] and International Towing Tank Conference(ITTC) instructions [8] proposed correction methods for the turning circle test; otherwise, sea trials should be implemented in deep, calm and non-restricted waters. The IMO, SNAME and ITTC rules solved the current direction, based on the uniform current hypothesis and the steady turning hypothesis [6,7,8]. However, the influence of wind load on ballast ships, container ships and ro-ro ships is greater than that on full-load oil tankers and bulk carriers. Moreover, the wind coefficient changes with wind direction, indicating that the wind load of the ship motion in turning circle is not constant; therefore, it cannot be regarded as the influence of uniform current. Thus, the installation of instruments and trial environments on sea trial vessels increase the difficulty of identification modeling, and thus the measurement data need to be corrected.

Compared with the ship model test, the sea trial has certain disadvantages which require improvement. Currently, the naval surface warfare center of America has a maneuvering and seakeeping tank to study ship motion in various sea conditions [9]. The National Maritime Research Institute (NMRI) established an actual sea model basin, using wind load and wave load simulation instruments [10,11], to research the performance of a full-scale ship in an actual sea. The indoor model test is organized, operated and validated by a professional organization and equipped with sophisticated towing devices and Charge Coupled Device(CCD) cameras; meanwhile, outdoor trials use high-precision satellite positioning instruments and shore-based wireless positioning devices on the sea. Due to the standardization and diversification of the test, the basin model test data quality is better than the real ship trial. Therefore, compared with basin model test, it is necessary to further process the data of the full-scale ship in order to improve the quality of its data.

In terms of wind force, Isherwood, Blendermann et al. and Fujiwara et al. used wind tunnel test data to fit the wind coefficients [12,13,14]: firstly, Isherwood proposed the estimation method for the calculation of wind force by formula, then Blendermann and Fujiwara updated the wind force formula structure and coefficients with a new wind tunnel experiment. Currently, the shipping industry focuses on the wind coefficient of container ships with dynamic stowage [15]. Aiming to calculate added mass, Motora proposed a simple method [16] and Zhou reproduced the formula for easy application [17]. For wave disturbance, Daidola used the second-order wave drift force and moment coefficient [18], Li used Daidola’s method for ship motion simulation [19]. Yasukawa studied the numerical prediction of second-order wave drift force [20], and Zhang et al. [21] and Hong et al. [22] studied second-order wave force and wave added resistance. Mei et al. [23] established a ship maneuvering model for a basin test; this paper will further explore actual sea ship maneuver modeling.

The paper is organized as follows: Section 2 briefly introduces the traditional methods that have been used to correct the full-scale sea trial. Section 3 proposes the novel correction method by wind, wave and sea surface current calculation. Section 4 explains the reference model and support vector machine (RM-SVM) for maneuvering modeling. Section 5 presents the case of trial correction. Section 6 presents maneuver modeling. Section 7 discusses the results of trial correction and motion modeling, and presents possible options for future works. Finally, Section 8 concludes this paper.

## 2. Traditional Correction Method

As shown in Figure 1, the literature [6,7,8] proposed a fast and convenient correction method called traditional correction method. In Figure 1, the blue line represents the turning circle track in a calm environment, the red line represents the turning circle track with disturbances and the green arrow represents the drift vector. The calm water track is a corrected −35° turning circle of a ship called Yukun, while the disturbed track is reproduced by one uniform surface current. The uniform surface current consisted of an east current, 0.5 m/s, and a north current, 0.5 m/s.

The SNAME [7] requires that the course changing of the turning circle is greater than 540°. It is assumed that the ship reaches a steady turning stage after 360°, and a steady drift velocity can be obtained by using the position data of the steady turning. As shown in the Figure 1, according to the last point of the track, the ship takes 250 s to drift 250 m in the east direction and drift 250 m in the north direction.

The correction process of Figure 1 is shown as the following: Suppose ship position as $\left(x_{i},y_{i}\right)$ and ship heading angle as $\psi_{i}$ at time $t_{i}$, and i∈{1,2,…,n}. Suppose ship position as $\left(x_{i}^{\prime },y_{i}^{\prime }\right)$ and ship heading angle as $\psi_{i}^{\prime }$ at time $t_{i}^{\prime }$, and $\psi_{i}$ − $\psi_{i}^{\prime }$ equals 360° or −360 °. In Figure 1, n is 4. Then, the drift distance between $\left(x_{i},y_{i}\right)$ and $\left(x_{i}^{\prime },y_{i}^{\prime }\right)$ is $l_{Di}$. The average drift velocity $\Delta l_{Di}$ between $\left(x_{i},y_{i}\right)$ and $\left(x_{i}^{\prime },y_{i}^{\prime }\right)$ can be estimated by following:(1)$$\Delta l_{Di}=\frac{\sum_{i=1}^{n}l_{Di}}{\sum_{i=1}^{n}\left(t_{i}-t_{i}^{\prime }\right)}=\frac{\sum_{i=1}^{n}{\left\|\left(x_{i},y_{i}\right)-\left(x_{i}^{\prime },y_{i}^{\prime }\right)\right\|}_{2}}{\sum_{i=1}^{n}\left(t_{i}-t_{i}^{\prime }\right)}$$

In the recommended process from IMO, SNAME and ITTC [6,7,8], the corrections were completed based on the assumption of uniform current and steady turning. The influence of a uniform current on a ship track increases linearly and is time-constant. However, the ship drifts induced by wind and waves are related to wind and wave direction angle, and the ship drifts are nonlinear and time-varying. Therefore, it is assumed that wind and wave disturbances are treated as linear; the nonlinear components are ignored. In this paper, the influences of wind, waves and currents are calculated separately on the basis of the hypothesis, and the improved method is proposed.

As this paper focus on maneuvering motion, the rolling, pitching and heaving of the ship are ignored by following explanations. Firstly, the drifts induced by wind and waves are treated as long term motion; meanwhile this manuscript focus on maneuvering motion, that is, only surge, sway and turning will be concerned. The rolling, pitching and heaving of the ship are therefore ignored. Secondly, the maneuvering is simplified as three degrees of freedom motion and is independent of seakeeping. The periodic motion of seakeeping has little effect on maneuvering motion with a large rudder angle. Thirdly, in the measurements, the rolling, pitching and heaving of the ship are periodic; thus, the motion, being periodic, can be filtered. Therefore, the maneuvering data can be used for correction and modelling.

## 3. Improved Correction Method

The improved correction method is mainly divided into three parts: first, calculate the wind force; then, calculate the wave drift force; finally, calculate the wind and wave drift distance. In this section, the surge and sway are corrected. Yaw is considered for the following reasons: Firstly, the ship hull underwater and the ship superstructure overwater, together, is close to being a box-shape. Thus, the yawing induced by wind and waves is negligible. Secondly, the yaw should be corrected for high precision; however, this will be much more complicated; this is because the yaw makes the heading angle change, and the heading angle changes the surge distance and sway distance.

### 3.1. Wind Load Calculation

Suppose that ship begins turning at time $t_{0}$. At time t, the ship velocity is V(t), heading angle is ψ(t), true wind velocity is $V_{T}\left(t\right)$, true wind direction is $\psi_{T}\left(t\right)$, the frontal wind load is $X_{w}\left(t\right)$ and lateral wind load is $Y_{w}\left(t\right)$. According to the reference [24], the wind force and its components, the force of the earth-centered earth-fixed (ECEF) east and ECEF north, change along with the ship heading. Thus, from $t_{0}$ to t, the wind induced drift distance in the earth-centered earth-fixed(ECEF)reference frame are $\Delta x_{w}\left(t\right)$ and $\Delta y_{w}\left(t\right)$, respectively, and are calculated as following:(2)$$\left\{ \begin{array}{c} \frac{\Delta x_{w}\left(t\right)}{\rho_{a}U_{R}^{2}\left(t\right)}=\int_{t_{0}}^{t}\int_{t_{0}}^{t}\left[\frac{A_{\mathrm{fw}}C_{\mathrm{wx}}\left(\alpha_{\mathrm{wR}}\left(t\right)\right)}{2m+2m_{x}}dt\cos\psi \left(t\right)-\frac{A_{\mathrm{lw}}C_{\mathrm{wy}}\left(\alpha_{\mathrm{wR}}\left(t\right)\right)}{2m+2m_{y}}dt\sin\psi \left(t\right)\right]dt\\ \frac{\Delta y_{w}\left(t\right)}{\rho_{a}U_{R}^{2}\left(t\right)}=\int_{t_{0}}^{t}\int_{t_{0}}^{t}\left[\frac{A_{\mathrm{fw}}C_{\mathrm{wx}}\left(\alpha_{\mathrm{wR}}\left(t\right)\right)}{2m+2m_{x}}dt\sin\psi \left(t\right)+\frac{A_{\mathrm{lw}}C_{\mathrm{wy}}\left(\alpha_{\mathrm{wR}}\left(t\right)\right)}{2m+2m_{y}}dt\cos\psi \left(t\right)\right]dt \end{array}\right.$$
where m is ship mass, $m_{x}$ and $m_{y}$ are added mass, $A_{\mathrm{fw}}$ and $A_{\mathrm{lw}}$ are the ship front projected area and lateral projected area, respectively, and $C_{\mathrm{wx}}$ and $C_{\mathrm{wy}}$ are wind coefficients of the ship front and lateral projected area, respectively. $U_{R}$ and $\alpha_{\mathrm{wR}}$ are relative wind velocity and direction, and can be calculated by $\psi_{T}$,$V_{T}$,V and ψ. Currently, the wind tunnel test is still the best means to determine the wind coefficient. Due to limited test facilities and high cost, the empirical formula of Blendermann [13] is applied in this paper. The added mass is calculated by the formulas from reference [16,17].

### 3.2. Wave Drift Force Calculation

Suppose that the wave drift force of ship longitude is $X_{d}\left(t\right)$ and wave drift force of ship transverse is $Y_{d}\left(t\right)$. According to reference [22], the second-order wave drift can be divided into ECEF (earth-centered earth-fixed) east and ECEF north. Thus, from $t_{0}$ to t, the wave drift force induced drift distance in ECEF reference frame are $\Delta x_{d}\left(t\right)$ and $\Delta y_{d}\left(t\right)$, respectively, and are calculated as following:(3)$$\left\{ \begin{array}{c} \Delta x_{d}\left(t\right)=\int_{t_{0}}^{t}\int_{t_{0}}^{t}\frac{X_{d}\left(t\right)}{m+m_{x}}dt\cos\psi \left(t\right)dt-\int_{t_{0}}^{t}\int_{t_{0}}^{t}\frac{Y_{d}\left(t\right)}{m+m_{y}}dt\sin\psi \left(t\right)dt\\ \Delta y_{d}\left(t\right)=\int_{t_{0}}^{t}\int_{t_{0}}^{t}\frac{X_{d}(t)}{m+m_{x}}dt\sin\psi \left(t\right)dt+\int_{t_{0}}^{t}\int_{t_{0}}^{t}\frac{Y_{d}\left(t\right)}{m+m_{y}}dt\cos\psi \left(t\right)dt \end{array}\right.$$

Due to the dynamic changing of the encounter frequency, the equivalent incident wave lengths $\lambda_{\mathrm{BX}}$ and $\lambda_{\mathrm{BY}}$ are introduced and satisfy the following equation (Equation (4)). The equivalent incident wave length has been used in reference [22].
(4)$$\{\begin{array}{c} E_{X}=X_{d}\lambda_{\mathrm{BX}}=X_{d}\frac{\lambda }{-\cos\alpha_{d}}\\ E_{Y}=Y_{d}\lambda_{\mathrm{BY}}=Y_{d}\frac{\lambda }{-\cos\alpha_{d}} \end{array}$$
where λ is incident mean wave length of the sea area and $\alpha_{d}$ is the wave direction. 

For real-time requirements, the Daidola formula [18] is used to calculate second-order wave drift force. The Daidola method has been applied in reference [19]. The surge and sway second-order wave drift force $X_{d}\left(t\right)$ and $Y_{d}\left(t\right)$ are as the following:(5)$$\left\{ \begin{array}{l} X_{d}\left(t\right)=\frac{\rho gL^{2}\zeta^{2}}{2}\left[0.05-0.2\left(\frac{\lambda_{\mathrm{BX}}}{L}\right)+0.75{\left(\frac{\lambda_{\mathrm{BX}}}{L}\right)}^{2}-0.51{\left(\frac{\lambda_{\mathrm{BX}}}{L}\right)}^{3}\right]\cos\left(\alpha_{d}\right)\\ Y_{d}\left(t\right)=\frac{\rho gL^{2}\zeta^{2}}{2}\left[0.46+6.83\left(\frac{\lambda_{\mathrm{BY}}}{L}\right)-15.65{\left(\frac{\lambda_{\mathrm{BY}}}{L}\right)}^{2}+8.44{\left(\frac{\lambda_{\mathrm{BY}}}{L}\right)}^{3}\right]\sin\left(\alpha_{d}\right) \end{array}\right.$$
where ζ is the mean wave height of the sea area and ζ, λ and $\alpha_{d}$ are calculated by wind velocity, based on the hypothesis of the fully developed wave and the hypothesis of long-crested wave. Therefore, the mean wave height of the sea area and mean wave length were estimated by wind force and direction information.

### 3.3. Resultant Distance Induced by Wind, Wave and Current

Based on wind load calculation and wave drift force calculation, the resultant distance induced by the wind, waves and currents is calculated as following:(6)$$\{\begin{array}{c} \Delta x\left(t\right)=k_{1}\Delta x_{w}\left(t\right)+k_{3}\Delta x_{d}\left(t\right)+k_{5}\Delta x_{c}\left(t\right)\\ \Delta y\left(t\right)=k_{2}\Delta y_{w}\left(t\right)+k_{4}\Delta y_{d}\left(t\right)+k_{6}\Delta y_{c}\left(t\right) \end{array}$$
where $k_{1}$~$k_{6}$ are adjusting parameters and $\Delta x_{c}$ and $\Delta y_{c}$ are east current and south current set as 1.0 m/s.

Suppose during time $\left[t_{1},t_{2}\right]$, the ship position is $\left(x\left(t_{j}\right),y\left(t_{j}\right)\right)$,$t_{j}\in \left[t_{1},t_{2}\right]$. Then, the correction ship position $\left(\hat{x}\left(t_{j}\right),\hat{y}\left(t_{j}\right)\right)$ can be calculated as following:(7)$$\{\begin{array}{c} \hat{x}\left(t_{j}\right)=x\left(t_{j}\right)-\Delta x\left(t_{j}\right)\\ \hat{y}\left(t_{k}\right)=y\left(t_{k}\right)-\Delta y\left(t_{j}\right) \end{array}$$

Based on the hypothesis of steady turning, the following equation has a solution for adjusting parameters $k_{i}$:(8)$$\begin{array}{l} \underset{k_{i}|i=1,2,3,4,5,6}{\arg\min}\sum_{t_{j}=t_{1}}^{t_{2}}{\left\|\left(\hat{x}\left(t_{j}\right),\hat{y}\left(t_{j}\right)\right)-\left(x_{0},y_{0}\right)\right\|}_{2}\\ \mathrm{subject}:\left\{ \begin{array}{l} \left(x_{0},y_{0}\right)=f_{C}\left(\hat{x}\left(t_{j}\right),\hat{y}\left(t_{j}\right)\right)\\ k_{i}\in [-10,+10] \end{array}\right. \end{array}$$
where $\left(x_{0},y_{0}\right)$ is center of a circle and can be solved by $f_{C}$; $f_{C}$ is a function from Kasa [25]. Meanwhile, $k_{i}\in [-10,+10]$ is a restrict condition for abnormal current. The value of these coefficients, $k_{i}$, are estimated by the optimization algorithm called pattern search. This function is established based on steady turning. The steady turning is a hypothesis condition from the IMO, ITTC and SNAME methods. Based on this hypothesis, the correction will form the final stage of turning in a circle.

To sum up this section, the illustration is shown in Figure 2. Figure 2 introduces the drift distances induced by the wind, waves and sea surface currents. The distances are divided into their east and north components. This distances also consist of the total drift distance in order to correct the ship track.

At the end of this section, based on the corrected ship position $\left(\hat{x}\left(t_{j}\right),\hat{y}\left(t_{j}\right)\right)$ and heading angle ψ(t), the velocities of surge, sway and yaw are derived. These velocities are called identified ship velocities, and are written as $u_{T}$, $v_{T}$ and $r_{T}$, where “T” stands for identified ship.

## 4. Maneuver Modeling Method

In this section, the RM-SVM whole ship model is established. Firstly, the RM-SVM model is identified with trial data. Based on the prediction of RM-SVM, the acceleration data and velocity data are reproduced. Then, the data is used to identify whole ship model by the least square algorithm.

The data from RM-SVM is smooth and does not have noise. These advantages will make the least square result much more precise than the trials data. In addition, the hydrodynamic coefficients are stable and not over-fitting.

### 4.1. RM-SVM Model

In this section, the corrected sea trial data is applied for ship maneuver modeling. This identification modeling method determines maneuverability aspects at rough sea and poor weather conditions, which is an important function used in order avoid collisions at actual sea conditions [26]. As introduced by Mei et al. [23], the reference model support vector machine (RM-SVM) method is utilized. Although modeling cannot describe wake information, as shown by Niu et al. [27], the model prediction precision is outstanding. Taking surge acceleration as example, suppose there existed an n identified ship measurement sample. The kth sample surge, sway and yaw velocity are $u_{T}\left(k\right)$, $v_{T}\left(k\right)$ and $r_{T}\left(k\right)$, and the kth surge acceleration function is $H_{T}\left(u_{T}\left(k\right),v_{T}\left(k\right),r_{T}\left(k\right),\delta_{T}\left(k\right)\right)$. For the RM, the kth sample surge, sway and yaw velocity are $u_{R}\left(k\right)$, $v_{R}\left(k\right)$ and $r_{R}\left(k\right)$, and the kth surge acceleration function is $H_{R}\left(u_{R}\left(k\right),v_{R}\left(k\right),r_{R}\left(k\right),\delta_{R}\left(k\right)\right)$, where “R” stands for reference model. The concept and selection method of the reference model is introduced in reference [23]. Based on the identified ship trials sample and reference model, the surge SVM can be written as the following:(9)$$L_{D}=\sum_{k=1}^{n}\left({\tilde{\alpha }}_{k}-\alpha_{k}\right)\Delta H\left(k\right)-\frac{1}{2}\sum_{k=1}^{n}\sum_{\ell =1}^{n}\left({\tilde{\alpha }}_{k}-\alpha_{k}\right)\left({\tilde{\alpha }}_{\ell }-\alpha_{\ell }\right){W_{k}}^{T}W_{\ell }-\varepsilon \sum_{k=1}^{n}\left({\tilde{\alpha }}_{k}+\alpha_{k}\right),$$
subject to:(10)$$\{\begin{array}{l} \sum_{k=1}^{n}\left(\alpha_{k}-{\tilde{\alpha }}_{k}\right)=0\\ \begin{array}{l} 0\leq \alpha_{k},{\tilde{\alpha }}_{k}\leq \tau \\ \alpha_{k}\left[\xi_{k}+\varepsilon -w^{T}W_{k}-l_{1}+\Delta H\left(k\right)\right]=0\\ {\tilde{\alpha }}_{k}\left[{\tilde{\xi }}_{k}+\varepsilon +w^{T}W_{k}+l_{1}-\Delta H\left(k\right)\right]=0\\ \alpha_{k}{\tilde{\alpha }}_{k}=0,\xi_{k}{\tilde{\xi }}_{k}=0\\ \left(\tau -\alpha_{k}\right)\xi_{k}=0,\left(\tau -{\tilde{\alpha }}_{k}\right){\tilde{\xi }}_{k}=0 \end{array} \end{array}$$
where ℓ=1,2,⋯,n is the order of sample data, α, $\tilde{\alpha }$, θ and $\tilde{\theta }$ are the Lagrangian multiplier vector of SVM hyper-plane, ξ and $\tilde{\xi }$ are the slack variable vector of SVM hyper-plane, w is the normal vector of SVM hyper-plane, $l_{1}$ is constant bias of SVM hyper-plane, τ is the regularization constant and ε is the Insensitive-band parameter. $W_{k}$ is the SVM input vector, as following:(11)$$W_{k}={\left(u_{T}\left(k\right),v_{T}\left(k\right),r_{T}\left(k\right),\delta_{T}\left(k\right)\right)}^{T}$$

Substituting sample data into Equations (9) and (11), the surge SVM is solved.

In the same way as the surge SVM, the sway and yaw SVM can be calculated. In addition, the identified ship accelerations can be predicted as the following:(12)$$\{\begin{array}{l} {\dot{u}}_{T}\left(t\right)={\dot{u}}_{R}\left(t\right)+w^{T}{\left(u_{T}\left(t\right),v_{T}\left(t\right),r_{T}\left(t\right),\delta_{T}\left(t\right)\right)}^{T}+l_{1}\\ {\dot{v}}_{T}\left(t\right)={\dot{v}}_{R}\left(t\right)+p^{T}{\left(v_{T}\left(t\right),r_{T}\left(t\right),{\dot{v}}_{T}\left(t\right),{\dot{r}}_{T}\left(t\right),\delta_{T}\left(t\right)\right)}^{T}+l_{2}\\ {\dot{r}}_{T}\left(t\right)={\dot{r}}_{R}\left(t\right)+q^{T}{\left(v_{T}\left(t\right),r_{T}\left(t\right),{\dot{v}}_{T}\left(t\right),{\dot{r}}_{T}\left(t\right),\delta_{T}\left(t\right)\right)}^{T}+l_{3} \end{array}$$
where ${\dot{u}}_{T}$, ${\dot{v}}_{T}$ and ${\dot{r}}_{T}$ are the identified ship sway and yaw accelerations, ${\dot{u}}_{R}$, ${\dot{v}}_{R}$ and ${\dot{r}}_{R}$ are the RM sway and yaw accelerations, p and q are the normal vector of sway and yaw SVM hyper-plane and $l_{2}$ and $l_{3}$ are constant bias of sway and yaw SVM hyper-plane, respectively. The Equation (12) can be solved by Runge–Kutta integration.

### 4.2. Whole Ship Model

Based on the prediction of Equation (12), the identified ship accelerations ${\dot{u}}_{T}$, ${\dot{v}}_{T}$ and ${\dot{r}}_{T}$ are reproduced by the RM-SVM model. Following that the input vector is [$u_{T}$, $v_{T}$, $r_{T}$], the output is [${\dot{u}}_{T}$, ${\dot{v}}_{T}$, ${\dot{r}}_{T}$]. Once the input vector and output vector are submitted into Equation (13), the whole ship model from reference [28] is identified with the least square method. The whole model structure and parameters are list as Equation (13).
(13)$$\{\begin{array}{l} \begin{array}{l} \left(m^{\prime }-X_{\dot{u}}^{\prime }\right){\dot{u}}_{T}^{\prime }=X_{\eta }^{\prime }(1-\eta_{T})+X_{\eta \eta }^{\prime }{\left(1-\eta_{T}\right)}^{2}+X_{\eta \eta \eta }^{\prime }{\left(1-\eta_{T}\right)}^{3}+X_{vv}^{\prime }{v_{T}^{\prime }}^{2}\\ +(X_{rr}^{\prime }+m^{\prime }x_{G}^{\prime }){r_{T}^{\prime }}^{2}+X_{\delta \delta }^{\prime }{\delta_{T}^{\prime }}^{2}+(X_{vr}^{\prime }+m^{\prime })v_{T}^{\prime }r_{T}^{\prime }+X_{vv\eta }^{\prime }{v_{T}^{\prime }}^{2}(1-\eta_{T})+X_{\delta \delta \eta }^{\prime }{\delta_{T}^{\prime }}^{2}(1-\eta_{T}) \end{array}\\ \begin{array}{l} \left(m^{\prime }-Y_{\dot{v}}^{\prime }\right){\dot{v}}_{T}+\left(m^{\prime }x_{G}^{\prime }-Y_{\dot{r}}^{\prime }\right){\dot{r}}_{T}=Y_{0}^{\prime }+Y_{v}^{\prime }v_{T}^{\prime }+Y_{vvv}^{\prime }{v_{T}^{\prime }}^{3}+Y_{vrr}^{\prime }v_{T}^{\prime }{r_{T}^{\prime }}^{2}+(Y_{r}^{\prime }-m^{\prime })r_{T}^{\prime }+Y_{rrr}^{\prime }{r_{T}^{\prime }}^{3}\\ +Y_{vvr}^{\prime }{v_{T}^{\prime }}^{2}r_{T}^{\prime }+Y_{\delta }^{\prime }\delta_{T}^{\prime }+Y_{\delta \delta \delta }^{\prime }{\delta_{T}^{\prime }}^{3}+Y_{\eta }^{\prime }(1-\eta_{T})+Y_{\eta \eta }^{\prime }{\left(1-\eta_{T}\right)}^{2}+Y_{\delta \eta }^{\prime }\delta_{T}^{\prime }(1-\eta_{T})+Y_{\delta \eta \eta }^{\prime }\delta_{T}^{\prime }{\left(1-\eta_{T}\right)}^{2} \end{array}\\ \begin{array}{l} \left(m^{\prime }x_{G}^{\prime }-N_{\dot{v}}^{\prime }\right){\dot{v}}_{T}^{\prime }+\left(I_{z}^{\prime }-N_{\dot{r}}^{\prime }\right){\dot{r}}_{T}^{\prime }=N_{0}^{\prime }+N_{v}^{\prime }v_{T}^{\prime }+N_{vvv}^{\prime }{v_{T}^{\prime }}^{3}+N_{vrr}^{\prime }v_{T}^{\prime }{r_{T}^{\prime }}^{2}+(N_{r}^{\prime }-m^{\prime }x_{G}^{\prime })r_{T}^{\prime }+N_{rrr}^{\prime }{r_{T}^{\prime }}^{3}\\ +N_{vvr}^{\prime }{v_{T}^{\prime }}^{2}r_{T}^{\prime }+N_{\delta }^{\prime }\delta_{T}^{\prime }+N_{\delta \delta \delta }^{\prime }{\delta_{T}^{\prime }}^{3}+N_{\eta }^{\prime }(1-\eta_{T})+N_{\eta \eta }^{\prime }{\left(1-\eta_{T}\right)}^{2}+N_{\delta \eta }^{\prime }\delta_{T}^{\prime }(1-\eta_{T})+N_{\delta \eta \eta }^{\prime }\delta_{T}^{\prime }{\left(1-\eta_{T}\right)}^{2} \end{array} \end{array}$$
where the $\eta_{T}=u_{T}/u_{0T}$, $u_{0T}$ is the ship service speed.

## 5. The Case of Trial Correction

In this section, the improved trial correction is applied to calculate the influences of wind, waves and currents. In addition, the drift distance and velocity of the turning circle test are solved. Then, the trial track and velocity for the full-scale ship are corrected.

### 5.1. General Details of Sea Trial

The study object of this paper is a motor vessel called Yukun; Table 1 and Figure 3 note Yukun particulars. The sea trial time was from 08:00 to 14:00 on 24 August 2012. The sea trial site is located in the northwest of the Yellow Sea, about 14 nautical miles from Dalian Port. The sea trials were carried out in open and deep water in clear and well weather conditions, as shown in Figure 4.

Figure 3 is part material from Dalian Maritime University and has been published in references [29,30].

From Figure 4, the maneuvers tests are listed in Table 2. 

As shown in Figure 5, the wind measuring system, differential global positioning system (DGPS), fiber-optic gyro and speed log are installed on the mast, bridge, gyro deck and ship bow, respectively. The DGPS position has a higher data update frequency than an automatic identification system [31].

### 5.2. Wind Load and Wave Drift Force Results

The time history subjected to the wind load and wave drift force for the +20° turning circle test were solved by the improved method of Equation (2). As shown in Figure 6, the surge and sway forces induced by wind and waves are calculated. The wind load shows dynamic fluctuations changing with time. As shown in Figure 5, the wind measuring system is shielded by the mast. Therefore, wind fluctuations included mast shielding, gusty components and random wind components. The details of the fluctuations also enhance the judgment of wind force and direction. 

In the calculation of Figure 6, the ship front projected area, $A_{\mathrm{fw}}$, is 297 m^2^, and lateral projected area, $A_{\mathrm{lw}}$, is 1304.6 m^2^. The longitude centroid position of $A_{\mathrm{lw}}$ is 2.46 m and the vertical centroid position is 6.8 m.

### 5.3. Wind- and Wave-Induced Acceleration Results

As shown in Figure 7, the surge and sway accelerations induced by wind and waves are calculated by Equation (5). 

### 5.4. Wind- and Wave-Induced Distance Results

As shown in Figure 8, the time history of Yukun being subjected to the wind- and wave-induced distance in the +20° turning circle test are solved by the improved method of Equations (6) and (8), respectively. The results of adjusting parameters $k_{1}$~$k_{6}$ are −2.46, 1.02, −1.24, 1, −0.21 and −0.37, respectively.

Figure 8 depicts the drift distance components induced by wind, waves and currents. The distances present the same order of magnitude of the wind, wave and sea surface current influence, none of distance can be ignored. The summery drift distances will be used to correct the ship track and calculate the surge sway and yaw velocities.

### 5.5. Track and Velocity Correction Results

The wind- and wave-induced distances have been used to correct the ship track; the comparison of the original turning circle and the corrected turning circle are presented in Figure 9. Figure 9 shows that the original turning circle moves significantly in the ECEF negative direction when under the influence of the wind and wave currents.

As shown in Figure 10, the surge sway and yaw velocities are corrected. As the turning reaches a steady stage, the corrected longitudinal velocity decreases and gradually converges the stable value. However, the log velocity increases at the stage of 200 s–250 s. The uncorrected sway velocity is stable at 0 m/s, while the corrected sway velocity increased rapidly, within 0 s–60 s, and converges to −1.52 m/s at 57 s. Since the yaw velocity is not corrected, the lines overlap.

All of the trials in Figure 4 and Table 2 have been corrected. In the previous manuscript, we only selected the 20° zigzag test for correction and modelling. Currently, the others are present in the Appendix B. These corrected cases of trials indicate that the improved correction method is valid for sea trials. These sea trials will be used for modelling in the next section.

## 6. The Case of Maneuver Modeling

In this section, the ship maneuvering model is established by the zigzag tests 6, 7, 8 and 9 from Figure 4. Identification model is one data driven-based method, and it is a common method in the maritime field [32]. In addition, the prediction model of +20° turning circle test is trained by Equations (9)–(11). Based on the method proposed in reference [22], the S175 ship is selected as the reference model of Yukun. In addition, the SVM is trained by zigzag test. Therefore, the RM-SVM of Yukun maneuvering model is established.

### 6.1. +20° Turning Circle Test

The +20° turning circle test is predicted by Equation (12), as shown in Figure 11.

The following conclusions can be drawn from Figure 9. According to the overall prediction results, the values of RM-SVM are relatively stable, without significant numerical anomalies and fluctuations, which indicates the stability of the identification model established by RM-SVM. According to the velocity prediction results, the values of RM-SVM are close to the corrected values. Since the yaw moment is not considered in the solution of influence, the yaw velocity is not corrected, so the corrected yaw velocity is equal to the original value.

### 6.2. The Ship Hydrodynamic Coefficient Result and −35° Turning Circle Validation

Based on the Equation (12), the RM-SVM model is acquired. Then, the RM-SVM is used to produce the accelerations data [${\dot{u}}_{T}$, ${\dot{v}}_{T}$ and ${\dot{r}}_{T}$], velocities data [$u_{T}$, $v_{T}$, $r_{T}$] and rudder angle data $\delta_{T}$. These data consist of input and output samples. Using the least-square linear regression algorithm and submitting the sample into Equation (13), the ship hydrodynamic coefficients are estimated, as shown in Table 3. The details of ship hydrodynamic coefficients are noted in [28].

Based on the hydrodynamic coefficients in Table 3, the −35° turning circle is predicted. The ship mass and inertia moment are known. The added mass and added moment are estimated by reference [3,16,17] as $m^{\prime }-X_{\dot{u}}^{\prime }=0.010249$, $m^{\prime }-Y_{\dot{v}}^{\prime }=0.017853$, $m^{\prime }x_{G}^{\prime }-Y_{\dot{r}}^{\prime }=0.00071412$, $m^{\prime }x_{G}^{\prime }-N_{\dot{v}}^{\prime }=0.001086$ and $I_{z}^{\prime }-N_{\dot{r}}^{\prime }=0.000015514$. The −35° turning circle test prediction results are as shown in Table 4.

As presented in the Table 4, the advance from the sea trial and prediction by RM-SVM are both smaller than the limit of the IMO standard for ship maneuverability 4.5*L*, as is the tactical diameter 5*L*. It is found that the full-scale ship complies with the IMO standard. On the other hand, the advance prediction accuracy of RM-SVM is 93% of the sea trial result, and the tactical diameter is 97%. This accuracy shows the high precision of the maneuver modeling.

## 7. Discussion

For the correction method, the improved method calculates the impact of the wind, waves and currents, but the traditional method takes wind and waves as a uniform current. Thus, the improved method proposed in this paper is a general form, whereas the traditional method is a special form. However, this does not mean that the improved one is perfect. Generally speaking, sea trial requires a buoy or radar wave system to measure the wave height. The Yukun test does not have this device. As described in the methodology, the improved method supposes that the wind and waves have been fully developed, and the sea state and induced ship motion are taken as stationary processes. The wave height and wave length are predicted via wind force. On the other hand, in the calculation of ship drift, the yaw induced by wind and wave has been ignored. Therefore, future works may consider correcting the yaw on the actual sea. 

For the maneuvering modeling part, the ship motion with the constant engine setting is predicted, and the precision is good. The engine setting condition of the sea trials satisfies the IMO standard for ship maneuverability. However, as in the Maritime Autonomous Surface Ship (MASS), the requirement of ship maneuvering will be much more technically demanding. The other conditions, such as engine RPM changing and ballasted loading, will be common in future research. It is foreseeable that the ship maneuver modeling will be associated with MASS for sophisticated ship path planning, tracking and collision avoidance.

## 8. Conclusions

In this paper, the measurement data of the installation equipment of the full-scale motor vessel were checked, wind and wave influences were solved and eliminated and sea trial track and velocity were corrected. Based on the corrected free running sea trials, the maneuvering model of the full-scale ship was established. Zigzag tests were used as training data to predict the turning circle test. Based on the identification model, the accelerations were reproduced. Finally, the whole ship model was identified and the modeling performance of +35 turning circle test was verified. To sum up the above work, the following conclusions can be drawn:(1)Due to the sea trial track and velocity being difficult to use for modeling directly, based on the assumption of the full developed wind and wave, an improved sea trial correction method was proposed. In this method, the wind, wave and current drift influences were calculated separately, and the adjusting parameters for the optical drift distances were solved by pattern search algorithm. The corrected track and velocity vectors were applied to modify the original data. The correction results of all trials illustrated the effectiveness of the proposed method.(2)According to the prediction example of the Yukun +20° turning circle test, it can be concluded that the maneuver model was precise. On the basis of the estimation results, the ship hydrodynamic coefficients in whole ship model were identifiable. From the track prediction of a −35° turning circle, the Yukun satisfies the IMO standard for ship maneuverability. In addition, the accuracy of the advance and tactical diameter reached 93% and 95%.(3)It will be much more convincing to validate this manuscript in several ships. However, it is not easy to obtain sea trials, as only the Yukun motor vessel test was organized and collected. In the future, there will be a new motor vessel built for maritime autonomous surface ship (MASS) research at Dalian Maritime University. The public building project has been approved. The correction and modeling of the new MASS will appear soon, once the trials are carried out.(4)Nowadays, ship maneuvering in waves is a tough and hot issue for the researcher as presented by ITTC 2017. Full-scale maneuvering in waves, including the rolling, heaving and pitch of ships, will be included in future works as soon as possible.

## Figures and Tables

**Figure 1 sensors-20-03963-f001:**
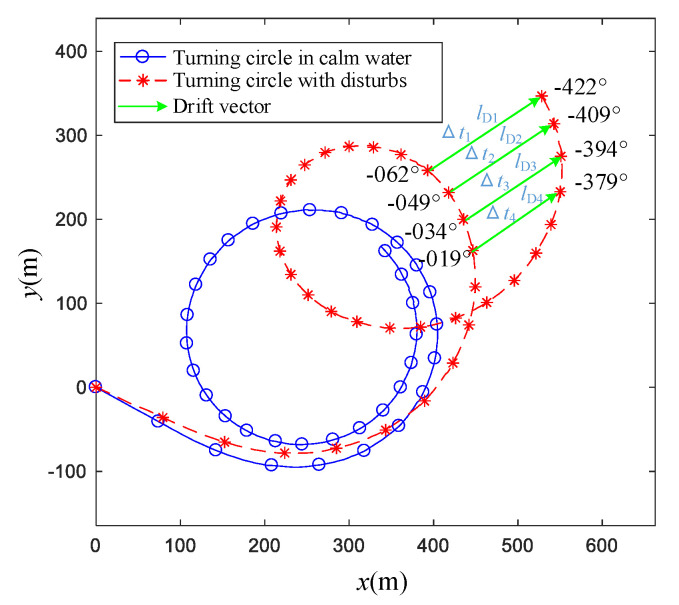
Traditional correction method of turning circle test.

**Figure 2 sensors-20-03963-f002:**
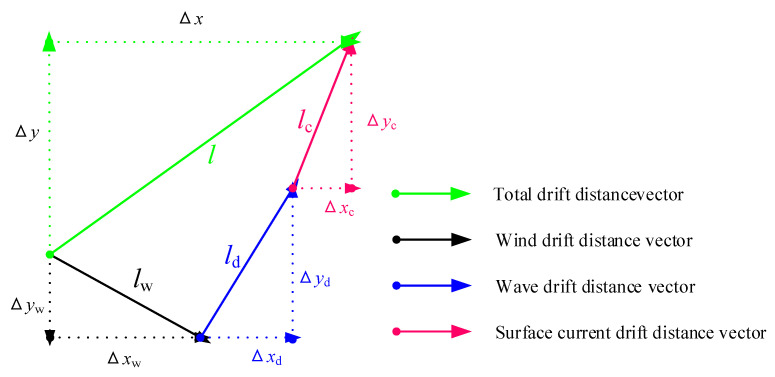
The drift distances and components induced by the wind, waves and surface currents.

**Figure 3 sensors-20-03963-f003:**
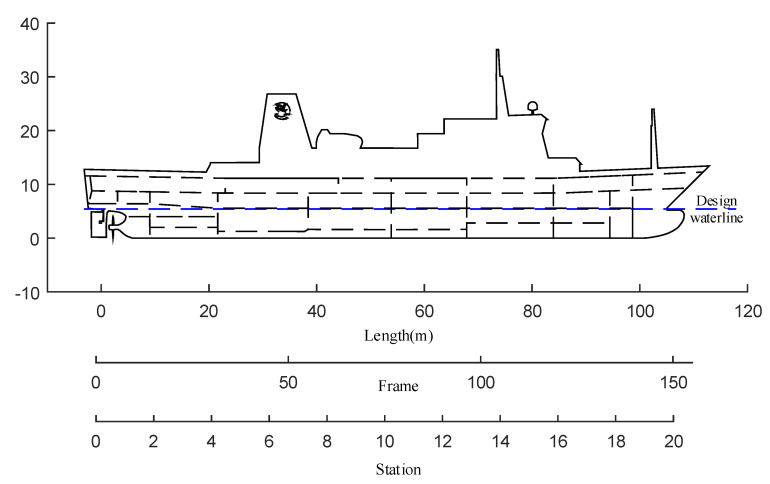
A general arrangement of Yukun motor vessel [29,30] (the figure permission has been achieved from Dalian Maritime University).

**Figure 4 sensors-20-03963-f004:**
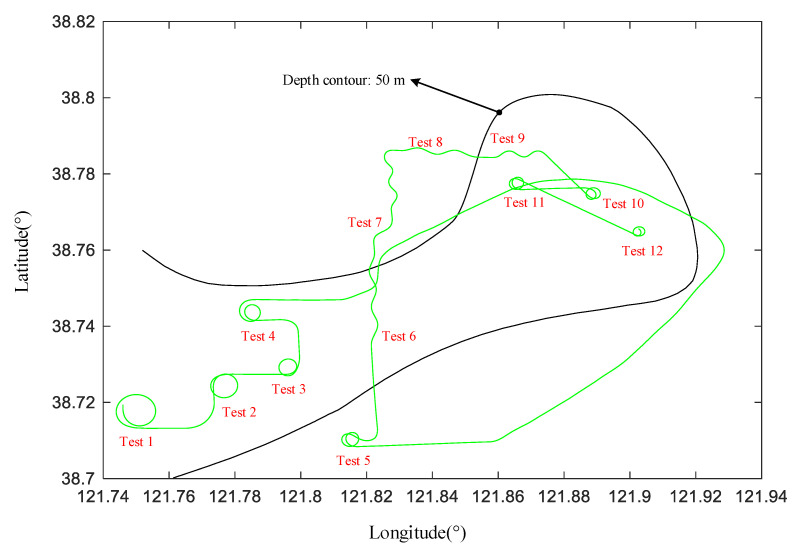
Sea trial area and water depth of **Yukun** test.

**Figure 5 sensors-20-03963-f005:**
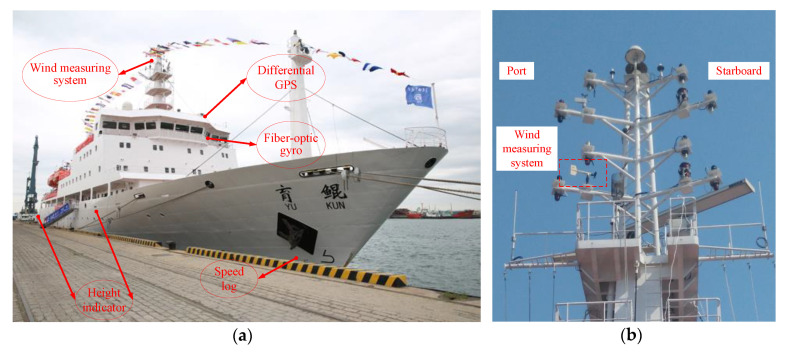
Instrument installation on board of Yukun. (**a**) The position, velocity and heading angle measuring instrument; (**b**) the wind measuring system installation on the mast.

**Figure 6 sensors-20-03963-f006:**
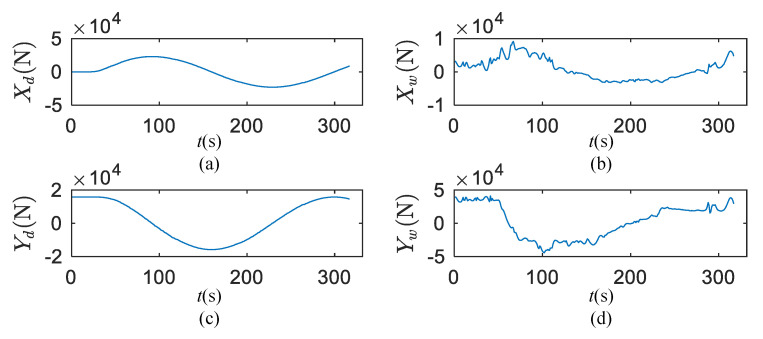
Wind load and wave drift force for +20°turning circle test. (**a**) Surge force induced by wave; (**b**) surge load induced by wind; (**c**) sway force induced by wave; (**d**) sway load induced by wind.

**Figure 7 sensors-20-03963-f007:**
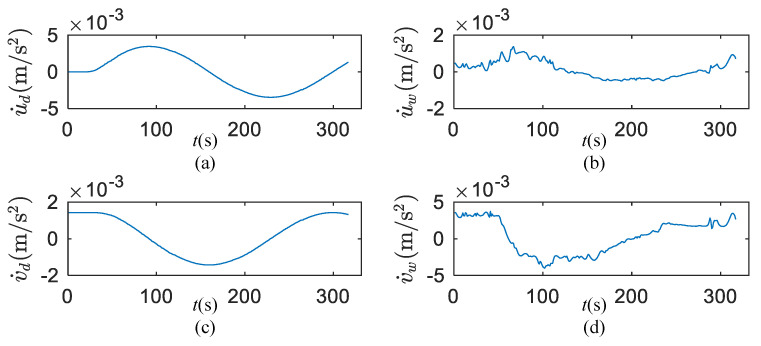
Wind- and wave-induced accelerations of +20° turning circle test. (**a**) Surge acceleration induced by wave; (**b**) surge acceleration induced by wind; (**c**) sway acceleration induced by wave; (**d**) sway acceleration induced by wind.

**Figure 8 sensors-20-03963-f008:**
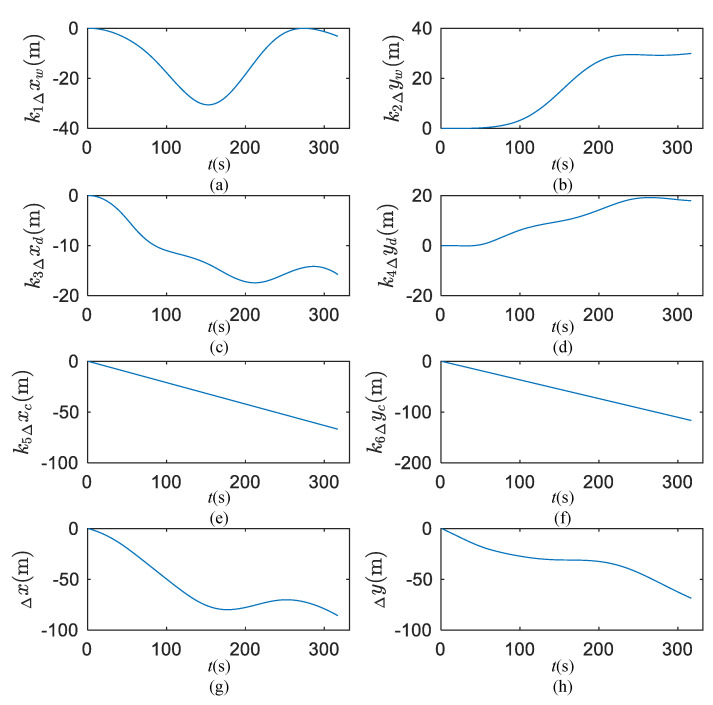
Wind, waves and sea surface currents induced earth-centered earth-fixed (ECEF) distances of +20° turning circle test. (**a**) Transverse drift distance induced by wind; (**b**) longitude drift distance induced by wind; (**c**) transverse drift distance induced by wave; (**d**) longitude drift distance induced by wave; (**e**) transverse drift distance induced by current; (**f**) longitude drift distance induced by current; (**g**) transverse resultant drift distance induced by wind, wave and current; (**h**) longitude resultant drift distance induced by wind, wave and current.

**Figure 9 sensors-20-03963-f009:**
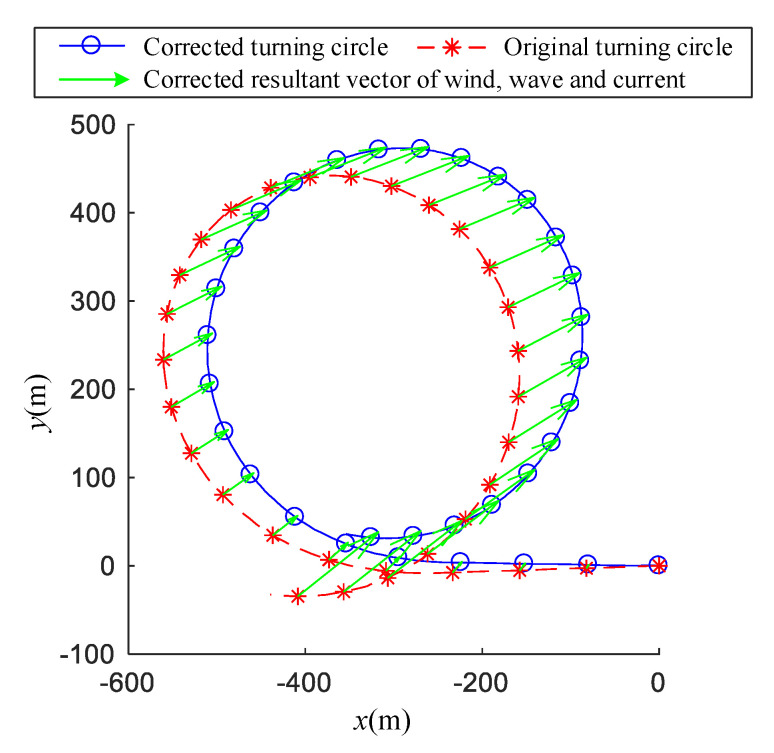
Correction of drift distance induced by the wind, waves and currents for +20° turning circle test.

**Figure 10 sensors-20-03963-f010:**
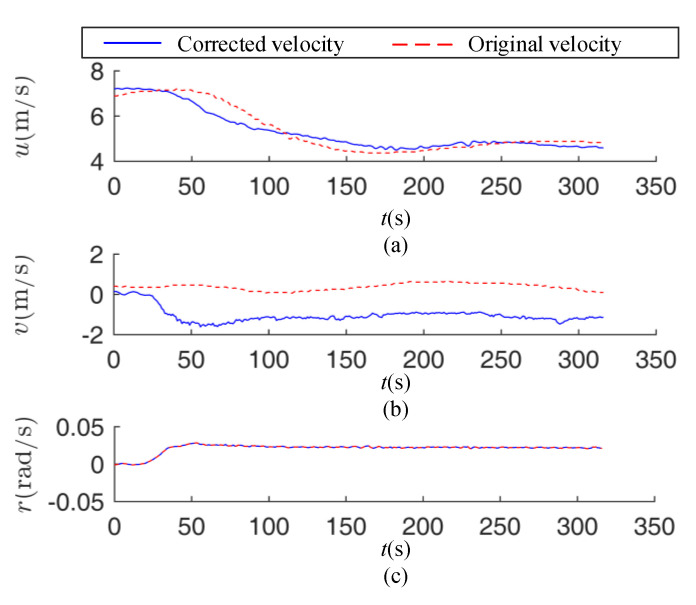
Correction of ECEF reference frame velocities for +20° turning circle test. (**a**) Ship surge velocity; (**b**) ship sway velocity; (**c**) ship yaw velocity.

**Figure 11 sensors-20-03963-f011:**
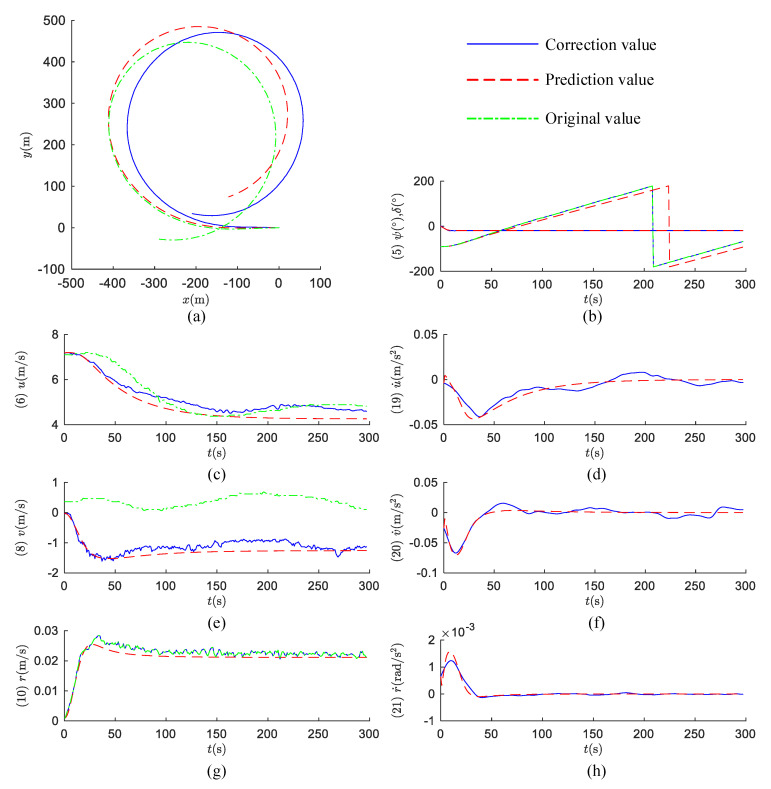
Prediction for +20° turning circle test. (**a**) Ship track; (**b**) ship heading angle and rudder angle; (**c**) surge velocity; (**d**) surge acceleration; (**e**) sway velocity; (**f**) sway acceleration; (**g**) yaw velocity; (**h**) yaw acceleration.

**Table 1 sensors-20-03963-t001:** Ship particulars of Yukun motor vessel.

Particulars	Values	Particulars	Values
Displacement	5710.2	Rudder area	11.8 m^2^
Length overall	116 m	Rudder height	4.8 m
Length between perpendiculars	105 m	Propeller diameter	3.8 m
Designed waterline length	106.5 m	Blade number	4
Ship breadth	18 m	Blade area ratio	0.67
Full-load draft	5.4 m	Maximum rudder rate	2.8°/s
Block coefficient	0.56	Prismatic coefficient	0.58

**Table 2 sensors-20-03963-t002:** Maneuver type and rudder angle details for all of the trials.

NO.	Time Points (s)	Maneuvers Type	Rudder Angle (°)	Sample Points
1	38213–38769	Turning circle	5	556
2	39232–39635	Turning circle	10	403
3	39898–40231	Turning circle	15	333
4	40619–40915	Turning circle	20	296
5	44577–44985	Turning circle	25	408
6	45509–45946	zigzag	10/−10	437
7	45975–46542	zigzag	20/−20	567
8	46606–46910	zigzag	10/−10	304
9	46999–47263	zigzag	20/−20	264
10	47517–47912	Turning circle	25	395
11	48178–48566	Turning circle	−30	388
12	49070–49450	Turning circle	34	380

**Table 3 sensors-20-03963-t003:** Identification results of non-dimensional surge, sway and yaw hydrodynamic coefficients for Yukun ship model.

$\mathrm{Surge}\;\mathrm{Coefficients}\;(\times 10^{5})$		$\mathrm{Sway}\;\mathrm{Coefficients}\;(\times 10^{5})$		$\mathrm{Yaw}\;\mathrm{Coefficients}\;(\times 10^{5})$	
$X_{\prime }^{\delta \delta }$	−130.4	$Y_{0}^{\prime }$	−69.1	$N_{0}^{\prime }$	−3.8
		$Y_{\delta }^{\prime }$	1079.1	$N_{\delta }^{\prime }$	−124.0
		$Y_{\delta \delta \delta }^{\prime }$	−808.7	$N_{\delta \delta \delta }^{\prime }$	104.3
$X_{\eta }^{\prime }$	630.3	$Y_{\eta }^{\prime }$	622.3	$N_{\eta }^{\prime }$	38.4
$X_{\eta \eta }^{\prime }$	−1637.8	$Y_{\eta \eta }^{\prime }$	−1397.1	$N_{\eta \eta }^{\prime }$	−80.0
$X_{\eta \eta \eta }^{\prime }$	3135.2	$Y_{\delta \eta }^{\prime }$	−1711.9	$N_{\delta \eta }^{\prime }$	194.2
$X_{\delta \delta \eta }^{\prime }$	−981.7	$Y_{\delta \eta \eta }^{\prime }$	5978.4	$N_{\delta \eta \eta }^{\prime }$	−670.6
$X_{vv}^{\prime }$	−1159.0	$Y_{v}^{\prime }$	283.8	$N_{v}^{\prime }$	−59.9
$X_{vv\eta }^{\prime }$	0.0	$Y_{vvv}^{\prime }$	−12,927.7	$N_{vvv}^{\prime }$	3647.3
$X_{rr}^{\prime }$	59.5	$Y_{r}^{\prime }-m^{\prime }$	249.2	$N_{r}^{\prime }-m^{\prime }x_{G}^{\prime }$	−53.8
		$Y_{rrr}^{\prime }$	−4829.9	$N_{rrr}^{\prime }$	418.3
$X_{vr}^{\prime }+m^{\prime }$	419.2	$Y_{vrr}^{\prime }$	−23,264.0	$N_{vrr}^{\prime }$	2774.4
		$Y_{vvr}^{\prime }$	−23,891.8	$N_{vvr}^{\prime }$	4561.3

**Table 4 sensors-20-03963-t004:** Validation for −35° turning circle test of Yukun full-scale ship and comparison with IMO standard.

Method		Advance	Tactical Diameter
**IMO standard for ship maneuverability**		4.5*L_PP_*	5*L**_PP_*
Sea trial result	Value	3.21*L_PP_*	2.98*L_PP_*
	Percentage	71%	60%
Prediction by RM-SVM in this paper	Value	2.99*L_PP_*	3.08*L_PP_*
	Percentage	66%	62%
	Prediction accuracy	93%	97%

Where *L_PP_* is ship length between perpendiculars.

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
