# Peer review of "Full-Scale Maneuvering Trials Correction and Motion Modelling Based on Actual Sea and Weather Conditions"

_sensors, 2020, doi:10.3390/s20143963_

Round 1
Reviewer 1 Report
Thank you for the paper. It would have been better that the calculations can cover other ships, since only one has been included, but obtain sea trials is not easy. Some comments / questions follow:
-I think it would be usefull to cite the first reference regarding wind forces on ships:
Isherwood, R. M., 1973. Wind resistance on merchant ships. Transactions of Rina, vol. 115.
- In how are the added masses computed?
- Classical seakeeping and manouvering theories uses crossed flow between motions, such as coupling 35 in heave and pitch, and 24 between sway and roll, but I thing you are not considering this, could you please explain this?
- Are you using adding inertias? I have not see them on the text. I guess you are using a pure 2D model without considering yaw.
- Wave height is not used in the wave drift force calculations?
- In line 124, please explain how you select the value of this coefficient
- There are some errata when referencing equations, i.e. line 140 a blank space is missing. Also in 145...
- A general arrangement figure of the ship next to table 1 will be useful
- Include also prismatic coefficient in table 1
- I see in Fig. 2 that several turning manoeuvres were made during the trial, but you are only presenting a few in the paper. I am not sure if you are limited in pages in this paper, but consider to reproduce more manoeuvres
- In 227, which formulation are you using for the hydrodynamic coefficients? consider to reproduce some equations there from reference [24]
- Consider to revise the English language by a native speaker.
Reviewer 2 Report
This manuscript considers the correction of the measurements of maneuverability parameters obtained from full-scale sea trials under disturbed sea and weather conditions. Major comments regarding this manuscript are listed below.
- The manuscript language needs revision and improvement starting from the title and the abstract. For example “Actual Sea Measurement” should be corrected to “Actual Sea and Weather Conditions”, “wind wave and current” to “wind, waves, and sea surface currents”, “ the ship becomes to be a steady turning” to “the ship reaches a steady turning state”, “The correction process of Figure 1 as following” does not have a verb, “the model prediction precise is outstanding” is grammatically wrong since “precise” is an adjective and it is technically confusing.
- The presented method aims at correcting maneuverability parameters measured during tests of full-scale sea-trials. In general, the sea trials are controlled tests which are carried under normal conditions. The true significance of such corrections are therefore not in sea trials but in determining maneuverability aspects at rough sea and weather conditions such as for risk identification, sea for example https://doi.org/10.3390/jmse8010005, where it is crucial to determine the vessel’s ability to stop or turn to avoid collision at actual sea conditions which are very different from the controlled test conditions. This important aspect is however not discussed in the literature review.
- Which data are used for Figure 1? What are the measurements for wind, waves, and surface currents in the disturbed case?
- From where were equations (2, 3) derived?, short explanation or references are missing there.
- Reference [189] is mentioned before equation (5), there is an error there.
- It is better to include an illustrative figure in section 3 to explain the equations and the symbols used therein.
- In section 4, what is the difference between subscripts “T” and “R”? what does the reference model stand for? In the scope of using SVM, these may refer to reference training data and testing data, but the full scale maneuvering trials are done only once.
- “Maneuvering Identification Modeling” is confusing. It should be maneuver prediction, modeling, or identification.
- It is not clear how section 4 relates to the manuscript, it seems isolated and its methodologies are not used.
- It is also recommended to re-write section 4. The SVM models, training data, test data, inputs, outputs, the purpose and the use of the models should be explained.
- The presented correction method is validated on one vessel and very few tests only, these are insufficient validation cases.
Round 2
Reviewer 1 Report
The revised version is much better that the orginal submission. I see that you have included a figure of ONLY the aft stations. Please, add full lines drawing of the ship, so your resulls could be validated numerically by other researchers.
Thank you
Reviewer 2 Report
This revision took all the comments into full consideration with very detailed and organized responses. The manuscript was improved and it is worth-publishing.
Figure 3 is of insufficient visibility quality, consider replacing the figure. The caption of the figure should refer to the original source as a reference. The permission should be obtained from copyright owners to use the figure. The permission statement needs to be mentioned in the caption of the figure.
However, if these cannot be made, it is OK to just remove figure 3.
